# Sliding Mode Flight Control Law Design Requirements for Oblique Wing Aircraft Based on Perturbation Theory

**Lixin Wang** [1], **Xun Sun** [1], **Hailiang Liu** [1], **Jingzhong Ma** [2], **Wenyuan Cheng** [3], **Shang Tai** [1,*], **Yun Zhu** [2] **and Ting Yue** [1,*]

1   School of Aeronautic Science and Engineering, Beihang University, Beijing 100191, China; wanglixin_buaa@163.com (L.W.); xunsun@buaa.edu.cn (X.S.); lhl_buaa@126.com (H.L.)
2   AVIC Jiangxi Hongdu Aviation Industry Group Company Ltd., Nanchang 330096, China; mjz_avic@126.com (J.M.); zy_avic@126.com (Y.Z.)
3   Aviation Industry Development Research Center of China, Beijing 100012, China; cwy_adr@126.com
*   Correspondence: taishang@buaa.edu.cn (S.T.); yueting_buaa@163.com (T.Y.)

**Abstract:** Flight control law parameters should be designed to provide a sufficient stability margin for closed-loop aircraft while ensuring command tracking accuracy. The singular perturbation margin (SPM) and generalized gain margin (GGM), which are generalizations of the classical phase margin (PM) and gain margin (GM), respectively, from a linear time-invariant system to a nonlinear time-varying system, can be used to quantitatively characterize the maximum singular perturbation and regular perturbation allowed to maintain system stability. In this paper, the sliding mode flight control structure and the design parameters of the sliding mode control law are first introduced for an oblique wing aircraft (OWA), the SPM-gauge and GGM-gauge are added to this closed-loop aircraft model, and the analytical expressions of the SPM and GGM are derived with respect to the control law parameters. Second, the stability margin design requirements of closed-loop aircraft in flight control system design specifications are converted into limitations on the SPM and GGM to determine the value range of the flight control law parameters. Then, with the goal of reducing the sum of the approaching time and sliding time, the parameter value combination is selected within the control law parameter range that meets the stability margin requirements, thus forming a flight control law design method for OWA during the wing skewing process. Finally, the designed control law parameters are applied to a sample OWA, and the stability margin of closed-loop aircraft during the wing skewing process is verified.

**Keywords:** stability margin; singular perturbation margin; generalized gain margin; sliding mode flight control; oblique wing aircraft

## 1. Introduction

An oblique wing aircraft (OWA) is an asymmetric layout aircraft that can rotate its wing at different flight velocities, forming various wing sweep configurations, with one side swept forwards and the other side swept back, as shown in Figure 1 [1]. During the oblique wing skewing process, the rotating wing will not only produce changes in lift, drag, and pitch moments but also generate asymmetric side forces, roll moments, and yaw moments. In addition, the wing components can have nonlinear interference with the fuselage, resulting in nonsteady time-varying aerodynamic forces during the wing skewing process. The variation rules of this time-varying aerodynamic force are complex, and it is impossible to accurately characterize the functional relationship between the time-varying aerodynamic force and the wing skewing rate. Therefore, the three-axis motion of the OWA during the wing skewing process has coupling and nonlinear characteristics, making it difficult to establish an accurate aerodynamic model, and resulting in uncertainty in closed-loop aircraft. Sliding mode control is a nonlinear control method that ensures the dynamic quality of a system through the design of a sliding mode function and reaching

law. It can generate control signals in a targeted manner based on changes in controlled system state variables, thereby causing the state variables to move along a specified sliding surface [2–5]. The flight control law designed using the sliding mode control method has a certain adaptability to changes in the configuration and aerodynamic parameters of OWA. In addition, the robustness of the sliding mode control is strong, and only a set of control law parameters are needed to eliminate the disturbances caused by aerodynamic modeling errors, noise measurement errors, and large-scale variations in aircraft motion parameters during the wing skewing process, without the need for frequent switching of the control law parameter values, making it more suitable for wing skewing processes with uncertainty in aerodynamic models.

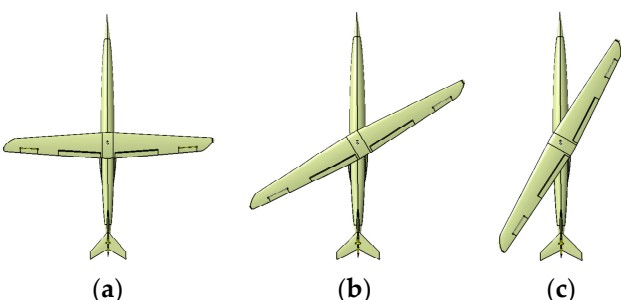

|  (a)  |  (b)  |  (c)  |

**Figure 1.** Layout of the oblique wing aircraft. (**a**) 0° skew angle; (**b**) 30° skew angle; (**c**) 60° skew angle.

The stability margin is an important criterion for evaluating whether the design values of flight control law parameters are suitable for real flight environments. As the stability metrics of linear time-invariant systems, the phase margin (PM) and gain margin (GM) have become the basic criteria for the design of flight control law parameters. The PM is defined as the angle difference between −180° and the open-loop frequency characteristics of the system with an open-loop gain of 1. It can reflect the maximum lag phase at which the system can still maintain stability for a specific frequency input signal with a gain of 1 [6]. The GM is defined as the inverse value of the open-loop amplitude–frequency characteristics of the system when there is a 180° phase lag in the feedback signal, reflecting the maximum range of open-loop gain changes allowed for the system to maintain stability [6]. However, the calculation of the PM and GM needs to be based on the linearized model of the control system. Due to the no-fixed configuration of OWA during the wing skewing process, the equilibrium point of closed-loop aircraft will change in real time with different skewing angles, resulting in the linear model of closed-loop aircraft not being unique. Even if a certain intermediate configuration in the skewing process is linearized, the analytical expression for the stability margin cannot be derived due to the high order and complexity of the linearized model. Therefore, it is difficult to quantify the impact of the control law parameter values on the stability of closed-loop aircraft using only the PM and GM.

In recent years, Yang X et al. [7–12] developed two new types of stability margin metrics based on perturbation theory for linear time-invariant, linear time-varying, and nonlinear systems, namely, the singular perturbation margin (SPM) and generalized gain margin (GGM). In control theory, perturbations are usually categorized into singular perturbations and regular perturbations. Singular perturbation refers to the perturbation that changes the nominal system's order, such as the transmission time delay of state feedback signals, and unmodelled elastic deformation of the wings and fuselage. Regular perturbation refers to perturbation that does not change the nominal system's order, such as the measurement errors in the state feedback signals, and changes in the aircraft mass and center of the gravity position. Under this classification condition, the SPM is defined as the maximum singular perturbation value allowed to maintain system stability. The GGM is defined as the maximum regular perturbation value allowed to maintain system stability [7]. In reference [7], the bijective correspondence between the SPM and PM, as

well as between the GGM and GM, was derived for a linear time-invariant system, proving the equivalence between the two new types of stability margins and classical stability metrics. References [8–12] used the Lyapunov stability analysis method to provide the calculation steps for the SPM and GGM when there are nonlinear or time-varying factors in the system. However, the above two stability margins defined by perturbation theory still lack engineering application, and the minimum stability margin requirements for designing nonlinear time-varying systems have not been summarized.

In the field of flight control design for variable configuration aircraft, researchers have focused on applying new nonlinear or intelligent control methods to control law design [1,13–23]. The values of the control law parameters are mostly determined based on time-domain simulation results and are continuously tested until the closed-loop aircraft achieves good command tracking characteristics without considering the limitations of the stability margin requirements. Cheng L et al. [13] designed an $\mathcal{L}_1$ adaptive control law based on dynamic inversion for the morphing process of a variable sweep aircraft and verified that the flight control law has satisfactory command tracking performance and strong robustness. Xu W et al. [19] designed a switching adaptive backstepping control law for the speed and altitude command tracking process of a variable sweep aircraft and proved the stability of the closed-loop aircraft using Lyapunov stability theory. Ting Y et al. [1] designed a sliding mode control law for the wing skewing process of an OWA, achieving trajectory stability during transitions between different configurations. However, the control law parameter values for completing the aforementioned variable sweep [13,17–19], variable V-tail [20], wing folding [21], wing telescoping [22], and wing skewing [1,23] are not unique. Although different parameter combinations can ensure the stability of closed-loop aircraft, they will have different effects on the stability margin, which can be used to determine whether the control law can ensure a smooth transition of the configuration in real flight environments.

The main innovation of this paper is to introduce the stability margin requirement of closed-loop aircraft into the design process of sliding mode flight control law parameters, which can help designers easily select a combination of control law parameters that can provide appropriate stability margin, fast response speed, and fast error convergence speed for closed-loop aircraft from numerous control law parameters that only meet the tracking accuracy of flight commands. This paper is based on the research results of perturbation theory and flight control design theory. The SPM and GGM are used as the stability metrics for a closed-loop aircraft model, and a parameter design requirement is proposed for the sliding mode flight control law of OWA that considers stability margin. First, the sliding mode flight control structure and the design parameters of the control law for OWA are introduced. The SPM-gauge and GGM-gauge are added to the closed-loop aircraft, and the analytical expressions of the SPM and GGM with respect to the sliding mode control law parameters are derived. Second, based on the quasi-steady assumption, a closed-loop aircraft with a low wing skew rate is regarded as a linear time-invariant system. The design requirements for PM and GM in the flight control system design specifications are transformed into the limiting requirements for the SPM and GGM, respectively. Therefore, the value range of the flight control law parameters is determined. Then, with the goal of reducing the sum of the sliding mode reaching motion time and sliding motion time, a parameter value combination with a faster command tracking speed within the control law parameter value range that meets the stability margin requirements is selected. Finally, the design process of the sliding mode flight control law is presented for a sample OWA, and the verification of the closed-loop aircraft stability margin is completed.

## 2. Sliding Mode Flight Control Structure of OWA

The sliding mode flight control structure adopted by a general form of OWA during the wing skewing process is shown in Figure 2 [1]:

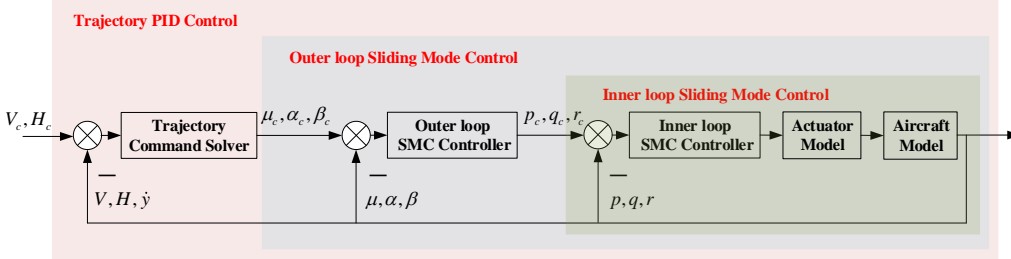

**Figure 2.** Sliding mode flight control structure of the OWA.

As shown in Figure 2, the flight control structure consists of three different functional control loops, namely a trajectory control loop, an outer sliding mode control loop, and an inner sliding mode control loop.

The trajectory control loop is used to maintain the stability of the OWA trajectory during the wing skewing process. The design of the trajectory command solver in Figure 2 adopts the PID control method, which can be used to solve the received flight speed command $V_c$ and flight altitude command $H_c$ into the engine thrust command $T_c$ and angle of attack command $\alpha_c$, respectively. At the same time, based on the feedback of the yaw displacement change rate $\dot{y}$, the roll angle command in the speed axis $\mu_c$ is generated, and together with $\alpha_c$, it serves as the input for the outer sliding mode control loop. Since the influences of the PID control law parameter values on the stability margin of closed-loop aircraft are not studied in this paper, the design process of trajectory command solver parameters is not discussed and its detailed description can be found in reference [1].

The outer sliding mode control loop is used to achieve attitude angle command tracking of the aircraft relative to the airflow coordinate system. The outer loop sliding mode controller can be used to calculate the received roll angle command in the speed axis $\mu_c$, angle of attack command $\alpha_c$, and side slip angle command $\beta_c$ into the roll rate command $p_c$, pitch rate command $q_c$, and yaw rate command $r_c$, which are used as inputs for the inner loop. As shown in Figure 2, the outer loop sliding mode control structure contains feedback for three state variables $p, q, r$. Therefore, it is necessary to design the corresponding sliding mode control law parameters for each of these three state variables.

The design of the outer loop sliding mode controller includes the design of the sliding surface $s$ and the design of the reaching law $\dot{s}$. The $s$ determines the motion equation of the closed-loop aircraft, while the $\dot{s}$ determines the output of the sliding mode controller. The design form of $s$ and $\dot{s}$ adopted in this paper is as follows [2,3]:

$$\begin{cases} s = e + c\int_0^t e\,dt \\ \dot{s} = -\Delta\mathrm{sgn}(s) - \omega s \end{cases} \tag{1}$$

where $e$ is the tracking error of the state variable on the command signal, $c > 0$ is the error integral gain, $\omega > 0$ is the reaching law index, $\Delta > 0$ is the approaching speed when the state variable reaches the sliding surface, and sgn( ) is the sign function. The value of parameter $\Delta$ can generally be chosen as a constant [24], so the parameters to be designed in Equation (1) are only $\omega$ and $c$.

For any control loop in state variable $\mu, \alpha, \beta$, the corresponding reaching law index and error integral gain need to be designed. Therefore, there are a total of 6 design parameters for the outer loop sliding mode controller: three reaching law indices $\omega_\mu, \omega_\alpha, \omega_\beta$ and three error integral gains $c_\mu, c_\alpha, c_\beta$.

The inner loop sliding mode control is used to track the three-axis angular velocity command of the aircraft. The inner loop sliding mode controller can be used to calculate the received three-axis angular velocity $p_c, q_c, r_c$ command as the required deflection angle command for each control surface, and these commands need to pass through the actuator before they can be used as inputs for the aircraft dynamics model. As shown in Figure 2,

the inner loop sliding mode control structure contains feedback for three state variables, $p,q,r$, and the corresponding sliding mode control law parameters need to be designed for each of these three state variables. The inner loop sliding mode controller is also based on the sliding mode surface and reaching law form in Equation (1), so the number of design parameters is also 6, namely, 3 reaching law indices $\omega_p, \omega_q, \omega_r$ and 3 error integral gains $c_p, c_q, c_r$.

In addition, the inner loop sliding mode control structure includes the dynamics model of the OWA. The dynamic model of an OWA is very different from that of a conventional fixed-wing aircraft. The characteristic of the wing rotating around the central axis determines that the OWA cannot be treated as a rigid body as in traditional modeling methods, but should be treated as a time-varying multibody dynamics system. The expression of its multibody dynamics model should include the influence of the wing skewing on the forces and moments. Common multi-body dynamics modeling methods include Newtonian mechanics modeling methods, Lagrange equation modeling methods, and Kane equation modeling methods. The vector-form dynamic equations used in modeling OWA in this paper are as follows [1,22]:

$$
\begin{cases}
\boldsymbol{F} = m\left(\dot{\boldsymbol{V}} + \boldsymbol{\omega} \times \boldsymbol{V}\right) + \frac{\delta\boldsymbol{\omega}}{\delta t} \times \boldsymbol{S} + 2\boldsymbol{\omega} \times \frac{\delta\boldsymbol{S}}{\delta t} + \boldsymbol{\omega} \times (\boldsymbol{\omega} \times \boldsymbol{S}) + \frac{\delta^2 \boldsymbol{S}}{\delta t^2} \\
\boldsymbol{M} = \boldsymbol{I} \cdot \frac{\delta\boldsymbol{\omega}}{\delta t} + \frac{\delta\boldsymbol{I}}{\delta t} \cdot \boldsymbol{\omega} + \boldsymbol{\omega} \times (\boldsymbol{I} \cdot \boldsymbol{\omega}) + \boldsymbol{S} \times \frac{\delta\boldsymbol{V}}{\delta t} + \boldsymbol{S} \times (\boldsymbol{\omega} \times \boldsymbol{V}) \\
\quad + \sum_{i=0}^{1} \left\{ \boldsymbol{I}_i \cdot \frac{\delta\boldsymbol{\omega}_i}{\delta t} + \frac{\delta\boldsymbol{I}_i}{\delta t} \cdot \boldsymbol{\omega}_i + \boldsymbol{\omega}_i \times (\boldsymbol{I} \cdot \boldsymbol{\omega}_i) + \frac{1}{m_i}\left[ \boldsymbol{S}_i \times \frac{\delta^2 \boldsymbol{S}_i}{\delta t^2} + \boldsymbol{\omega} \times \left( \boldsymbol{S}_i \times \frac{\delta\boldsymbol{S}_i}{\delta t} \right) \right] \right\}
\end{cases} \tag{2}
$$

where $\boldsymbol{F}$ and $\boldsymbol{M}$ are the forces and moments acting on the OWA, respectively. $m$ is the mass of the OWA. $\boldsymbol{V}$ is the flight speed. $\boldsymbol{\omega}$ is the angular velocity of rotation of the body axis system relative to the inertial coordinate system. $\boldsymbol{S}$ is the static moment of the OWA around the origin of the body axis system, representing the mass distribution of the aircraft, where the origin of the body axis system is located at the center of gravity of the OWA in the $0°$ skewing angle configuration. $i = 0$, 1 represents fixed fuselage and movable wing, respectively. $m_i$ is the rigid mass of each part of the OWA; $\boldsymbol{I}_i$ is the moment of inertia of each part relative to its own center of mass; $\boldsymbol{\omega}_i$ is the angular velocity of rotation of each part around the body axis system; $\delta\boldsymbol{S}_i/\delta t$ describes the change in mass distribution of the OWA relative to the body axis system caused by wing skewing.

The values of the reaching law index and error integral gain will affect the SPM and GGM of the control loop. It has been proven in references [25,26] that when multiple control loops in a system are perturbed simultaneously, as long as the perturbation value is less than the minimum stability margin among all control loops, the closed-loop system can still maintain stability. This conclusion indicates that the SPM and GGM of the closed-loop system are equal to the minimum SPM and GGM among all control loops, respectively. From a physical perspective, since the divergence of the response of any control loop can cause the divergence of the entire closed-loop system, when all control loops are subjected to the same perturbation, the loop with the lowest stability margin will first exhibit a divergence trend. Therefore, its stability margin can be used as the stability margin of the closed-loop system. To determine the SPM and GGM of the closed-loop aircraft, it is necessary to calculate the SPM and GGM of $\mu,\alpha,\beta,p,q,r$ control loops separately and then select the minimum value from them.

## 3. Relationship between the SPM and Sliding Mode Control Law Parameters

In this section, the control loop of the single-state variable is taken as an example and the Lyapunov stability analysis method is used to calculate the SPM of this loop to establish the bijective relationship between the SPM and the sliding mode control law parameters.

According to Figure 2, the closed-loop control structure of a single-state variable is as follows:

In Figure 3, $x$ is the state variable, which can be either the outer loop state variables $\mu,\alpha,\beta$ or the inner loop state variables $p,q,r$. $x_c$ is the command signal corresponding to the

state variable $x$. $e = x_c - x$ is the tracking error of the command signal. The sliding surface $s$ and reaching law $\dot{s}$ are based on the design form in Equation (1). The parameters to be designed are the reaching law index $\omega$ and error integral gain $c$.

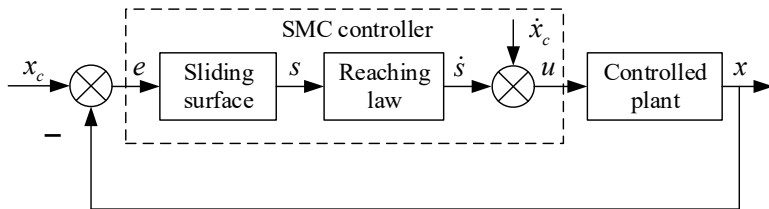

**Figure 3.** Closed-loop control structure with a single-state variable.

According to time-scale separation methods [1,22], the dynamic response of the inner loop fast variable is ignored in this paper when designing the outer loop; that is, considering that the inner loop fast variable has reached a steady state, the inner and outer loops can be separated for control law design. Therefore, when $x$ is selected as the outer loop state variable, the controlled plant in Figure 3 is a dynamic equation about $\mu,\alpha,\beta$. When $x$ is selected as the inner loop state variable, the controlled plant becomes the dynamic equation about $p,q,r$, and its specific form is detailed in reference [1].

The dynamic equation of the controlled plant can be written as a nonlinear affine form, as shown in Equation (3):

$$\dot{x} = F(x,t) + G(x)u \tag{3}$$

where $F(x,t)$ and $G(x)$ are the continuous functions related to the state variable. $u$ is the sliding mode control law. $t$ is the time. To maintain a stable flight path during the wing skewing process, reference [1] designed $u$ as follows:

$$u = G^{-1}(x)\left[-F(x,t) - \dot{s} + \dot{x}_c + ce\right] \tag{4}$$

This control law is related to the motion characteristics of the aircraft and the design parameters of the sliding mode controller and changes in real time with different flight states. By substituting Equation (4) into the dynamic Equation (3), the state equation of the system in Figure 3 can be obtained as follows:

$$\dot{x} = -\dot{s} + \dot{x}_c + ce = -\dot{s} + \dot{x}_c + c(x_c - x) \tag{5}$$

According to Equation (5), under the action of the sliding mode control law $u$, the motion characteristics of the closed-loop system depend only on the design form of the error integral gain $c$ and sliding mode reaching law $\dot{s}$ and are independent of the motion characteristics or scales of the controlled plant, that is, the forms of $F(x,t)$ and $G(x)$.

To impose singular perturbation on the closed-loop aircraft motion model, it is necessary to add subsystems with perturbation parameters to the structure in Figure 3. This subsystem is artificially introduced for calculating the SPM and is commonly referred to as the SPM-gauge [7]. The resulting singular perturbation system structure is shown in Figure 4.

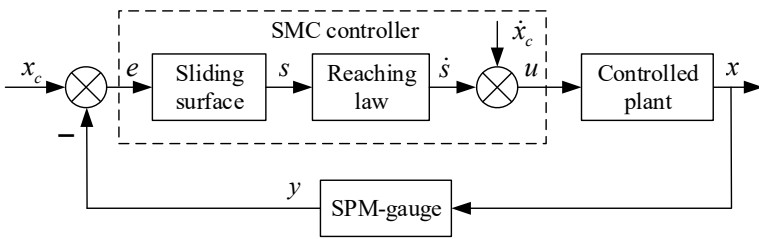

**Figure 4.** Singular perturbation structure of a single-state variable.

The method of imposing singular perturbation on the closed-loop aircraft in this paper is to increase the phase delay of the state variables in the feedback path. Therefore, the SPM-gauge in Figure 4 is selected as an all-pass filter with first-order response characteristics, and its state equation is expressed as follows [7]:

$$\begin{cases} \varepsilon\dot{z} = -\omega_0 z + 2\omega_0 x \\ y = z - x \end{cases} \tag{6}$$

In Equation (6), $\varepsilon$ is the singular perturbation parameter. $z$ is the state variable of the SPM-gauge. $y$ is the output of the SPM-gauge. $\omega_0$ is the filter constant, which affects the phase delay of the output signal $y$. Assuming that $\varepsilon \in (0,\varepsilon^*]$ is the range of singular perturbation parameter values that maintain the stability of the system in Figure 4, then $\varepsilon^*$ is an SPM.

By combining Equations (5) and (6), the state equation of the system in Figure 4 can be obtained as follows:

$$\begin{cases} \dot{x} = -\dot{s} + \dot{x}_c + c(x_c - y) \\ \varepsilon\dot{z} = -\omega_0 z + 2\omega_0 x \\ y = z - x \end{cases} \tag{7}$$

By letting $\left(\dot{x},\dot{z}\right) = (0,0)$, the equilibrium point of Equation (7) is determined to be $(x,z) = (x_c, 2x_c)$. To move this equilibrium point to the origin, variable substitutions of $\widetilde{e} = x - x_c$ and $\widetilde{y} = z - 2x_c$ are implemented, and Equation (7) is rewritten in the form of $\left(\widetilde{e},\widetilde{y}\right)$ as follows:

$$\begin{cases} \dot{\widetilde{e}} = -c\widetilde{e} - c\widetilde{y} - \dot{s} \\ \varepsilon\dot{\widetilde{y}} = -\omega_0\widetilde{y} - 2\varepsilon\left(-c\widetilde{e} - c\widetilde{y} - \dot{s}\right) \end{cases} \tag{8}$$

According to the sliding surface and reaching law designed in Equation (1), $\dot{s}$ in Equation (8) can be expressed as follows:

$$\dot{s} = -\Delta\mathrm{sgn}(s) - \omega\left(e + c\int_0^t e\,dt\right) = -\Delta\mathrm{sgn}(s) + \omega\widetilde{e} + \omega c\int_0^t \widetilde{e}\,dt \tag{9}$$

By setting $e_1 = \int_0^t \widetilde{e}\,dt$ and $e_2 = \widetilde{e}$ and substituting Equation (9) into Equation (8), the final form of the system state equation in Figure 4 can be obtained as follows:

$$\begin{cases} \dot{e}_1 = e_2 \\ \dot{e}_2 = -ce_2 - c\widetilde{y} + \Delta\mathrm{sgn}(s) - \omega e_2 - \omega c e_1 \\ \varepsilon\dot{\widetilde{y}} = -\omega_0\widetilde{y} - 2\varepsilon(-ce_2 - c\widetilde{y} + \Delta\mathrm{sgn}(s) - \omega e_2 - \omega c e_1) \end{cases} \tag{10}$$

The following is based on the Lyapunov stability analysis method to derive the range of singular perturbation parameters for maintaining System (10) stability.

Equation (11) is selected as an alternative Lyapunov function for System (10).

$$v = 0.5(\omega + c + 2\omega c)e_1^2 + e_1 e_2 + e_2^2 + 0.5\widetilde{y}^2 = 0.5[e_1, e_2]\begin{bmatrix} \omega + c + 2\omega c & 1 \\ 1 & 2 \end{bmatrix}\begin{bmatrix} e_1 \\ e_2 \end{bmatrix} + 0.5\widetilde{y}^2 \tag{11}$$

where $\omega + c + 2\omega c > 0.5$. Then, the derivative of $v$ along the system trajectory can be expressed as:

$$\dot{v} = (\omega + c + 2\omega c)e_1 e_2 + e_2^2 + e_1\dot{e}_2 + 2e_2\dot{e}_2 + \widetilde{y}\dot{\widetilde{y}} \tag{12}$$

where $e_1\dot{e}_2$, $e_2\dot{e}_2$ and $\widetilde{y}\dot{\widetilde{y}}$ satisfy the following inequality:

$$\begin{cases} e_1\dot{e}_2 \leq -ce_1e_2 - ce_1\widetilde{y} + \Delta|e_1| - \omega e_1e_2 - \omega ce_1^2 \\ \qquad \leq -(1-\theta_1)\omega ce_1^2 - (\omega+c)e_1e_2 - ce_1\widetilde{y}, \forall |e_1| \geq \frac{\Delta}{\omega c\theta_1} \\ e_2\dot{e}_2 \leq -ce_2^2 - ce_2\widetilde{y} + \Delta|e_2| - \omega e_2^2 - \omega ce_1e_2 \\ \qquad \leq -(1-\theta_2)(\omega+c)e_2^2 - \omega ce_1e_2 - ce_2\widetilde{y}, \forall |e_2| \geq \frac{\Delta}{(\omega+c)\theta_2} \\ \widetilde{y}\dot{\widetilde{y}} \leq -\frac{\omega_0}{\varepsilon}\widetilde{y}^2 + 2ce_2\widetilde{y} + 2c\widetilde{y}^2 + 2|y|\Delta + 2\omega e_2\widetilde{y} + 2\omega ce_1\widetilde{y} \\ \qquad \leq -(1-\theta_3)(\omega_0/\varepsilon - 2c)\widetilde{y}^2 + 2\omega ce_1\widetilde{y} + (2\omega + 2c)e_2\widetilde{y}, \forall |\widetilde{y}| \geq \frac{2\Delta}{(\omega_0/\varepsilon - 2c)\theta_3} \end{cases} \tag{13}$$

where $\theta_1, \theta_2, \theta_3 \in (0,1)$, whose values affect the final convergence accuracy of the state variables $e_1, e_2, \widetilde{y}$. To reduce the complexity of calculating the stability margin, $\theta_1, \theta_2, \theta_3$ can be taken as a constant. By substituting Inequality (13) into Equation (12), we determine that $\dot{v}$ satisfies the following inequality:

$$\dot{v} \leq -[|e_1|, |e_2|, |\widetilde{y}|] \begin{bmatrix} (1-\theta_1)\omega c & 0 & -\omega c + 0.5c \\ 0 & 2(1-\theta_2)(\omega+c) - 1 & -\omega \\ -\omega c + 0.5c & -\omega & (1-\theta_3)(\omega_0/\varepsilon - 2c) \end{bmatrix} \begin{bmatrix} |e_1| \\ |e_2| \\ |\widetilde{y}| \end{bmatrix} \tag{14}$$

When $\dot{v}$ is negative, the system in Figure 4 is stable. At this time, the quadratic coefficient matrix in Equation (14) must be positive definite. The first- to third-order principal minor determinants of the coefficient matrix are set to be positive, and $\theta_1 = \theta_2 = \theta_3 = 0.5$ is taken. Then, the range of $\varepsilon$ can be determined as follows:

$$0 < \varepsilon < \varepsilon^* = \frac{0.5\omega_0 m_1 m_3}{cm_1 m_3 + m_2^2 m_3 + m_4^2 m_1} \tag{15}$$

where $m_1 = 0.5\omega c$, $m_2 = -\omega c + 0.5c$, $m_3 = \omega + c - 1$, and $m_4 = -\omega$. The analytical expression for the SPM is given by Equation (15), which is mainly influenced by the filter constant $\omega_0$ and sliding mode control law parameters ($\omega,c$). Once the value of the singular perturbation parameter in the feedback loop is larger than $\varepsilon^*$, the motion of the closed-loop aircraft will exhibit a divergent trend.

## 4. Relationship between the GGM and Sliding Mode Control Law Parameters

In this section, the single-variable control loop of the OWA is still used as an example, and the Lyapunov stability analysis method is implemented to calculate the GGM of this loop to establish the bijective relationship between the GGM and the sliding mode control law parameters.

To impose regular perturbations on closed-loop aircraft, a subsystem with regular perturbation parameters, commonly referred to as a GGM-gauge [7], needs to be added to the structure of Figure 3. The resulting regular perturbation system structure is shown in Figure 5.

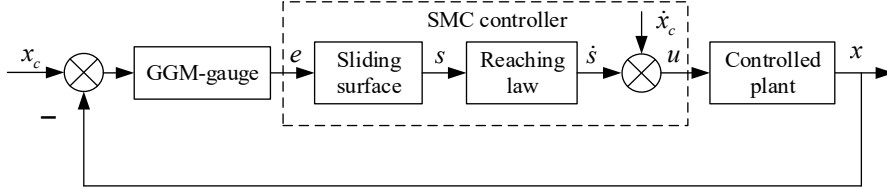

**Figure 5.** Regular perturbation structure of a single-state variable.

Since regular perturbation does not change the order of the original closed-loop aircraft model, the GGM-gauge takes the general form of a proportional component with a constant gain value $k$.

Regular perturbation on the closed-loop aircraft model is imposed in this paper by changing the tracking error of the command signal in the forward path. At this time, the

error signal received by the sliding mode controller changes from *e* to *ke*, and the control law *u* also changes from Equation (4) to Equation (16).

$$u = G^{-1}(x)\left[-F(x,t) - \dot{s} + \dot{x}_c + kc(x_c - x)\right] \tag{16}$$

By substituting Equation (16) into the aircraft dynamics in Equation (3), the motion form of the state variables when regular perturbation exists can be written as follows:

$$\dot{x} = -\dot{s} + \dot{x}_c + kc(x_c - x) \tag{17}$$

According to Equation (17), the motion characteristics of a regular perturbation system depend on the design of the regular perturbation parameter *k*, the error integral gain *c*, and the sliding mode reaching law $\dot{s}$ and are also independent of the motion characteristics or scales of the controlled plant.

By letting $\dot{x} = 0$, the equilibrium point of Equation (17) is determined to be $x = x_c$. To move this equilibrium point to the origin, a variable substitution of $\widetilde{e} = x - x_c$ is used, and Equation (17) is rewritten in the form of $\widetilde{e}$.

$$\dot{\widetilde{e}} = -\dot{s} - kc\widetilde{e} \tag{18}$$

According to the sliding surface and reaching law designed via Equation (1), the expression of $\dot{s}$ under regular perturbation can be obtained as follows:

$$\dot{s} = -\Delta\text{sgn}(ks) - \omega\left(ke + c\int_0^t ke\,dt\right) = -\Delta\text{sgn}(s) + \omega k\widetilde{e} + \omega kc\int_0^t \widetilde{e}\,dt \tag{19}$$

Letting $e_1 = \int_0^t \widetilde{e}\,dt$ and $e_2 = \widetilde{e}$, and then substituting Equation (19) into Equation (18), the state equation of the system in Figure 5 can be obtained as follows:

$$\begin{cases} \dot{e}_1 = e_2 \\ \dot{e}_2 = -\omega kce_1 - k(\omega + c)e_2 + \Delta\text{sgn}(s) \end{cases} \tag{20}$$

The following is based on the Lyapunov stability analysis method to derive the range of regular perturbation parameters for maintaining System (20) stability.

Equation (21) is selected as an alternative Lyapunov function for System (20).

$$v = 0.5e_1^2 + e_1e_2 + e_2^2 = 0.5[e_1, e_2]\begin{bmatrix} 1 & 1 \\ 1 & 2 \end{bmatrix}\begin{bmatrix} e_1 \\ e_2 \end{bmatrix} \tag{21}$$

Then, the derivative of *v* along the system's trajectory can be expressed as

$$\dot{v} = e_1\dot{e}_1 + \dot{e}_1e_2 + e_1\dot{e}_2 + 2e_2\dot{e}_2 \tag{22}$$

where $e_1\dot{e}_2$ and $e_2\dot{e}_2$ satisfy the following inequality:

$$\begin{cases} e_1\dot{e}_2 = -\omega kc \cdot e_1^2 - k(\omega + c)e_1e_2 + \Delta|e_1| \\ \quad \leq -(1 - \theta_4)\omega kc \cdot e_1^2 - k(\omega + c)e_1e_2, \forall|e_1| \geq \frac{\Delta}{\omega kc\theta_4} \\ e_2\dot{e}_2 = -k(\omega + c)e_2^2 - \omega kc \cdot e_1e_2 + \Delta|e_2| \\ \quad \leq -(1 - \theta_5)k(\omega + c)e_2^2 - \omega kc \cdot e_1e_2, \forall|e_2| \geq \frac{\Delta}{k(\omega + c)\theta_5} \end{cases} \tag{23}$$

where $\theta_4, \theta_5 \in (0,1)$, whose values affect the final convergence accuracy of the state variables $e_1, e_2$. To reduce the complexity of calculating the stability margin, $\theta_4$ and $\theta_5$ can be taken

as constants. By substituting the Inequality (23) into Equation (22), we determine that $\dot{v}$ satisfies the following inequality:

$$\dot{v} \leq -[|e_1|, |e_2|] \begin{bmatrix} (1-\theta_4)\omega kc & 0.5(k(\omega+c)+2\omega kc-1) \\ 0.5(k(\omega+c)+2\omega kc-1) & 2(1-\theta_5)k(\omega+c)-1 \end{bmatrix} \begin{bmatrix} |e_1| \\ |e_2| \end{bmatrix} \quad (24)$$

When $\dot{v}$ is negative, the system in Figure 5 is stable. At this time, the quadratic coefficient matrix in Equation (24) must be positive definite. The first- and second-order principal minor determinants of the coefficient matrix are set to be positive, and $\theta_4 = \theta_5 = 0.5$ is taken. Then, the inequality relationship satisfied by the regular perturbation parameter $k$ can be determined as follows:

$$(4n_1 - n_3)k^2 + (-4n_2 + 2n_3)k - 1 > 0 \quad (25)$$

where $n_1 = 0.5\omega c(\omega + c)$, $n_2 = 0.5\omega c$ and $n_3 = \omega + c + 2\omega c$. To place the solution of Inequality (25) within a bounded interval of positive real numbers, its quadratic coefficients and discriminant should satisfy the following expression:

$$\begin{cases} 4n_1 - n_3 < 0 \\ \Omega = (-4n_2 + 2n_3)^2 + 4(4n_1 - n_3) > 0 \end{cases} \quad (26)$$

The solution to inequality (26) is $k \in [k_{\min}, k_{\max}]$, where the expressions for $k_{\min}$ and $k_{\max}$ are as follows:

$$\begin{cases} k_{\min} = \frac{-(-4n_2 + 2n_3) + \sqrt{\Omega}}{2(4n_1 - n_3)} \\ k_{\max} = \frac{-(-4n_2 + 2n_3) - \sqrt{\Omega}}{2(4n_1 - n_3)} \end{cases} \quad (27)$$

Equation (27) provides the analytical expression of the generalized gain margins $k_{\min}$ and $k_{\max}$, which are influenced mainly by the parameters ($\omega$,$c$) of the sliding mode control law. Once the measurement value of the tracking error in the forward path is less than $k_{\min}$ times the nominal value or larger than $k_{\max}$ times the nominal value, the motion of closed-loop aircraft will exhibit a divergent trend.

## 5. Design Requirements for the Sliding Mode Control Law Parameters

### 5.1. Stability Margin Design Requirements for the Closed-Loop Aircraft Model

Wing skewing can cause changes in the flow field around an aircraft, which in turn generates nonlinear interference with fuselage components, resulting in complex unsteady aerodynamics. During the wing skewing process, the OWA is essentially a time-varying system. This is because there is relative motion between the wing and the fuselage, and configuration parameters such as the center of gravity position, moment of inertia, and product of inertia will change with the wing skewing angle. In terms of aerodynamic characteristics, wing skewing not only produces changes in lift, drag, and pitch moment but also produces side forces, roll moments, and yaw moments that are not present when the wing is symmetrical. These forces and moments will change in real time with the skewing angle and skewing rate.

However, there is currently a lack of mature and universal stability metrics for time-varying systems, and there is no reference that provides a design requirement for the stability margin of time-varying systems from theoretical or engineering practice. This paper does not study the case of time-varying systems, but based on the research results of reference [1], considers the OWA with a low wing skewing rate as a linear time-invariant system and uses the design specifications of linear time-invariant systems for stability margin to constrain the design values of the sliding mode control law parameters.

Reference [1] provides an unsteady aerodynamic modeling method for OWA, which is the quasi-steady assumption. This assumption ignores the unsteady aerodynamic forces caused by the wing skew rate and assumes that during the configuration changing process

with smaller skew rates, the aerodynamic forces of the OWA at each moment are equal to those of the static configuration under the same flight conditions. The resulting modeling errors can be eliminated through the robustness of sliding mode control. Based on the quasi-steady assumption, a closed-loop aircraft model with a low wing skew rate is considered in this paper as a linear time-invariant system. The design requirements for the PM and GM of the linear time-invariant system are transformed into limiting requirements for the SPM and GGM, respectively, as the basis for determining the values of the sliding mode control law parameters.

According to the design specification MIL-F-9490D for aircraft flight control systems, all control loops of linearized flight control systems should meet the following stability margin requirements [27]:

$$|PM| \geq 45° \qquad |GM| \geq 10dB \tag{28}$$

In Equation (28), the PM is the phase margin and the GM is the gain margin. PM > 0 represents the open-loop phase lag, and PM < 0 represents the open-loop phase lead. GM > 0 represents an increase in open-loop gain, while GM > 0 represents a decrease in open-loop gain. It has been proven in reference [7] that when SPM-gauge is selected as the first-order all-pass filter, as shown in Equation (6), the SPM and PM of a linear time-invariant system will have the following functional correspondence:

$$PM = \arctan\left(\frac{2\varepsilon^*}{1 - (\varepsilon^*)^2}\right), 0 < \varepsilon^* < 1 \tag{29}$$

Since the focus of this paper is only on the case of phase lag in feedback signals, by substituting PM $\geq 45°$ into Equation (29), the design requirements for the SPM in a linear time-invariant system can be obtained as follows:

$$0.414 \leq \varepsilon^* < 1 \tag{30}$$

By taking the design value of the filter constant $\omega_0$ a as 2 rad/s in Equation (6) [9] and setting $\varepsilon^* \geq 0.414$ in Equation (15), the range of values for the reaching law index and error integral gain that meets the requirements of SPM can be obtained, as shown in the region enclosed by the blue dashed line in Figure 6.

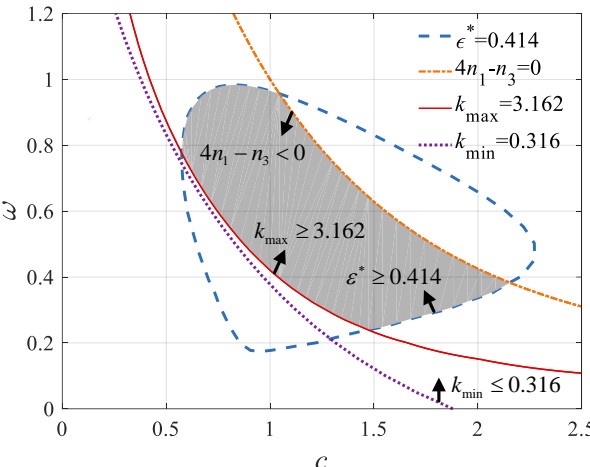

**Figure 6.** Value regions of the control law parameters that satisfy both SPM and GGM requirements.

In Figure 6, each point located on the blue dashed line represents a combination of control law parameter values that make the SPM of a single loop exactly equal to 0.414. The control law parameter values corresponding to each point inside the blue dashed line can increase the SPM of a single loop to greater than 0.414. When the values of the control

law parameters are outside the region enclosed by the blue dashed line, the SPM of a single loop will be smaller than 0.414.

The functional correspondence between the GGM and GM is expressed as follows [7]:

$$\begin{cases} 20\lg k_{\max} = |GM| \\ 20\lg k_{\min} = -|GM| \end{cases} \tag{31}$$

By substituting $|GM| \geq 10$ dB into Equation (31), the design requirements for the GGM of a linear time-invariant system can be obtained as follows:

$$\begin{cases} k_{\max} \geq 3.162 \\ 0 < k_{\min} \leq 0.316 \end{cases} \tag{32}$$

Since the perturbation parameters that satisfy the stability of the system in Figure 5 are located within the interval $k \in [k_{\min}, k_{\max}]$, according to Equation (32), this interval should include at least $[0.316, 3.162]$. This indicates that when the system in Figure 5 can continue to maintain stability, its open-loop gain can increase to at least 3.162 times the nominal value or decrease to at least 0.316 times the nominal value.

By substituting inequality (31) into the expression of GGM (27) and combining it with the prerequisite for the existence of a solution in Equation (26), the range of values for the reaching law index and the error integral gain that satisfies the requirements of GGM can be obtained, as shown in the common region among the purple dotted line, the red solid line, and the orange dash-dot line in Figure 6.

In Figure 6, since the curve $k_{\max} = 3.162$ is above the curve $k_{\min} = 0.316$, the control law parameter values that satisfy $k_{\max} \geq 3.162$ naturally satisfy $k_{\min} \leq 0.316$. Therefore, the range of control law parameter values obtained according to the requirements of the GGM is determined only by $k_{\max} \geq 3.162$ and the quadratic coefficient $4n_1 - n_3 < 0$.

The intersection of the four curves in Figure 6 is selected to obtain the range of sliding mode control law parameter values that meet both SPM and GGM requirements, as shown in the shaded area in Figure 6. The shaded area is enclosed by the curves $\varepsilon^* \geq 0.414$, $k_{\max} \geq 3.162$, and $4n_1 - n_3 < 0$. For the blue dashed boundary on both sides of this shaded area, the control law parameter corresponding to any point on it can make the SPM of a single loop exactly equal to 0.414 and the GGM $k_{\max}$ greater than 3.162. For the red solid line boundary, the control law parameter corresponding to any point on it can make the GGM $k_{\max}$ of a single loop exactly equal to 3.162 and the SPM greater than 0.414. For the orange dash-dot line boundary, the control law parameter corresponding to any point on it can make the quadratic coefficient $4n_1 - n_3$ in Equation (25) exactly equal to 0, and the value of the control law parameter needs to be below this orange dash-dot line to ensure that $4n_1 - n_3$ is strictly negative. Meanwhile, the control law parameter values near the lower part of this orange dash-dot line can make the SPM of a single loop greater than 0.414 and the GGM $k_{\max}$ greater than 3.162.

### 5.2. Parameter Values for the Sliding Mode Control Law

In this paper, the idea for determining the parameter values of the sliding mode flight control law is to accelerate the response speed and error convergence speed of closed-loop aircraft while ensuring stability margin design requirements and flight path stability.

The time-domain indicators of the sliding mode control law include the reaching time and sliding time, and the values of the control law parameters need to be determined based on their impacts on these two indicators. The reaching time $t_r$ refers to the time when the state variable first reaches the sliding surface from its initial position, representing the response speed of a system to commands. Sliding time $t_s$ refers to the time taken by a state variable from entering the sliding surface to converging to a steady-state value, reflecting a system's ability to regulate the response process. The following derives the influence of the reaching law index $\omega$ and the error integral gain $c$ on $t_r$ and $t_s$ as the design basis for the values value of $\omega$ and $c$.

According to Equation (1), the expression for the exponential reaching law is

$$\dot{s} = \begin{cases} -\Delta - \omega s & s > 0 \\ \Delta - \omega s & s < 0 \end{cases} \tag{33}$$

Equation (33) is subsequently solved to obtain the following:

$$s(t) = \begin{cases} (s_0 + \Delta)e^{-\omega t} - \Delta & s > 0 \\ (s_0 - \Delta)e^{-\omega t} + \Delta & s < 0 \end{cases} \tag{34}$$

where $s_0 = s(0)$. When the state variable reaches the sliding surface at time $t_r$, $s(t_r) = 0$. Therefore, letting $s(t) = 0$ in Equation (34), the reaching time $t_r$ is solved as follows:

$$t_r = t_r(\omega) = \begin{cases} \frac{1}{\omega}[\ln(\Delta + s_0) - \ln \Delta] & s > 0 \\ \frac{1}{\omega}[\ln(\Delta - s_0) - \ln \Delta] & s < 0 \end{cases} \tag{35}$$

According to Equation (35), increasing the value of $\omega$ can accelerate the reaching process and improve the system's tracking speed of command signals.

When the motion of the state variable remains on the sliding surface, $s(t) \equiv 0$. Therefore, by letting the sliding mode function in Equation (1) be zero, and the convergence law of the command tracking error can be obtained as follows:

$$e(t) = e(0) \cdot \exp(-ct) \tag{36}$$

Reference [28] suggested that when $e(t)/x_c \leq 1\%$, the control system achieved satisfactory command tracking performance. According to Equation (36), the sliding time is

$$t_s = t_s(c) = \frac{1}{c} \ln \frac{e(0)}{0.01 x_c} \tag{37}$$

To improve the response speed and error convergence speed of the closed-loop aircraft motion model, the total time $t(\omega, c) = t_r(\omega) + t_s(c)$ is taken as the objective function, and the value combination of parameter $(\omega, c)$ in the shaded area of Figure 6 is determined to reduce the value of $t(\omega, c)$.

The command $x_c$ is taken as a 1° step signal, and $x(0) = 0$ is assumed. Then, $e(0) = x_c - x(0) = 1°$, $s_0 = e(0) = 1°$. In addition, Equation (15) and Equation (27) show that the approaching speed $\Delta$ does not affect the SPM or GGM. In this paper, $\Delta$ is taken as a constant value of 0.5 [29], and the expression of the total time is obtained as follows:

$$t(\omega, c) = \frac{0.034}{\omega} + \frac{4.605}{c} \tag{38}$$

According to Equation (38), the total time $t(\omega, c)$ is inversely proportional to the values of $\omega$ and $c$, and the influence of the $c$ value on the total time is greater than that of $\omega$. Therefore, it is recommended to select the control law parameters on the right side of the shaded area in Figure 6, where the value of $\omega$ can be selected within the interval [0.24,0.64] and the value of $c$ can be selected within the interval [1.40,2.10].

## 6. Application of the Proposed Flight Control Law Design Requirements

### 6.1. The Sample OWA

In this section, an OWA is taken as an application example of the flight control law design requirements. The configuration parameters of this OWA at a 0° skew angle are shown in Table 1, and the control surface of the sample OWA is shown in Figure 7.

**Table 1.** Configuration parameters of the sample OWA.

| Parameters | Values | Unit |
|---|---|---|
| Takeoff weight | 13,000 | kg |
| Wingspan | 24.5 | m |
| Wing mean aerodynamic chord (MAC) | 2.0 | m |
| Wing area | 47.6 | $\text{m}^2$ |
| Roll axis moment of inertia | 100,075 | $\text{kg·m}^2$ |
| Pitch axis moment of inertia | 387,245 | $\text{kg·m}^2$ |
| Yaw axis moment of inertia | 287,238 | $\text{kg·m}^2$ |

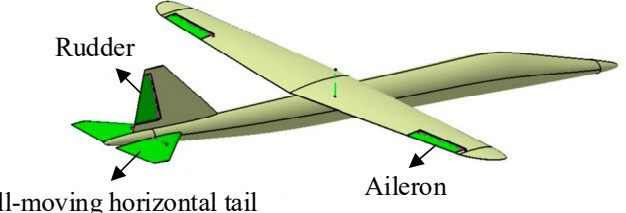

**Figure 7.** Configuration and control surface layout of the sample OWA.

The sample OWA is equipped with three sets of control surfaces: an all-moving horizontal tail, an aileron, and a rudder. The all-moving horizontal tail consists of two pieces on the left and right, with their deflection angles represented by $\delta_{eL}$ and $\delta_{eR}$, respectively. The left and right all-moving horizontal tails control the pitch motion of the aircraft through linkage deflection and assist rolling motion through differential deflection. An aileron is used for roll control, and its deflection angle is represented by $\delta_a$. A rudder is used for yaw control, and its deflection angle is represented by $\delta_r$.

The sample OWA adopts a first-order actuator model shown in Figure 8, which consists of the actuator bandwidth gain, deflection rate limit, integrator, and deflection angle limit.

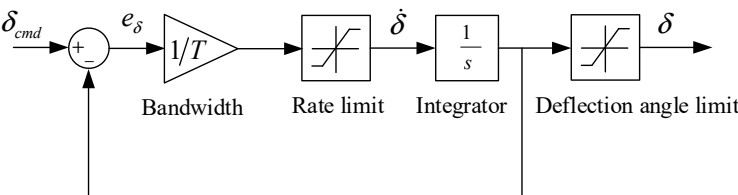

**Figure 8.** Actuator Model.

The transfer function of the actuator model is shown in Equation (39):

$$\delta = \frac{1}{Ts + 1}\delta_{cmd} \quad -\dot{\delta}_{\min} \leq \dot{\delta} \leq \dot{\delta}_{\max}, \quad -\delta_{\min} \leq \delta \leq \delta_{\max} \tag{39}$$

where $\delta_{\text{cmd}}$ is the input command for the actuator model. $1/T$ is the actuator bandwidth. $e_\delta$ is the tracking error of the actuator deflection angle. $\dot{\delta}_{\min}, \dot{\delta}_{\max}$ is the limit value of the actuator deflection rate; $\delta_{\min}, \delta_{\max}$ is the limit value of the actuator deflection angle, and $\delta$ is the output of the actuator model. The parameters of the actuator models are shown in Table 2.

**Table 2.** Parameters of the actuator model of the sample OWA.

| Parameters | All-Moving Horizontal Tail | Aileron | Rudder |
|---|---|---|---|
| Bandwidth | 20.2 rad/s | 20.2 rad/s | 20.2 rad/s |
| Rate limit | $[-80°/s, 80°/s]$ | $[-80°/s, 80°/s]$ | $[-80°/s, 80°/s]$ |
| Deflection angle limit | $[-25°, 25°]$ | $[-21.5°, 21.5°]$ | $[-30°, 30°]$ |

The sliding mode flight control structure of the sample OWA is shown in Figure 2. According to the analysis results in Chapter 2, this structure requires the design of 12 sliding mode control law parameters, which are six reaching law indices $\omega_i$ and six error integral gains $c_i$, $i = \mu,\alpha,\beta,p,q,r$.

According to the time-scale separation method in references [1,22], the dynamic response of the inner loop fast variable is ignored when designing the outer loop of the sample OWA. Therefore, the inner loop is regarded as the controlled plant of the outer loop, achieving the goal of designing the inner and outer loop control law parameters separately, as shown in Figure 9.

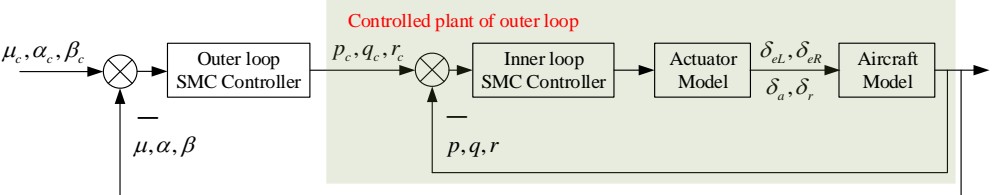

**Figure 9.** Treating the inner loop as the controlled plant of the outer loop.

As shown in Figure 9, when using the time-scale separation method for flight control design, the inner and outer loop sliding mode control structures are the same as the closed-loop control structure of a single-state variable in Figure 3. Therefore, in the control loops of the six state variables $\mu,\alpha,\beta$ and $p,q,r$, the SPM and GGM of any loop can be calculated using Equations (15) and (27).

According to the design requirements of the sliding mode control law parameters in Section 5.1, this paper selects the design points for the reaching law index and error integral gain as the intersection points of the blue dashed line and the red solid line in the lower right corner of Figure 6, that is, $(\omega, c) = (0.24, 1.47)$, corresponding to $\varepsilon^* = 0.414$, $k_{max} = 3.162$, and $k_{min} = 0.263$. On this basis, the design values of the six reaching law indices are all set to $\omega_i = 0.24$, and the design values of the six error integral gains are all set to $c_i = 1.47$, $i = \mu,\alpha,\beta,p,q,r$. Because the control loops of the six state variables in the inner and outer loops all adopt the same control law parameter values, according to Equations (15) and (27), the six control loops should have the same theoretical value of the stability margin, and the stability margin of the closed-loop aircraft should be equal to the stability margin of any of these control loops, also $\varepsilon^* = 0.414$, $k_{max} = 3.162$ and $k_{min} = 0.263$.

In the following subsection, singular and regular perturbations are imposed on the sample OWA through mathematical simulation, and the designed reaching law index and error integral gain are examined to determine whether they can meet the stability margin design requirements in Equations (30) and (32).

### 6.2. Verification of SPM and GGM

To verify the SPM and GGM of the closed-loop aircraft shown in Figure 2, considering the most severe case of disturbance imposition, six SPM-gauges, as shown in Equation (6), are added to the $p,q,r$ and $\mu,\alpha,\beta$ feedback paths of the inner and outer sliding mode control loops, and six GGM-gauges are added to the $p,q,r$ and $\mu,\alpha,\beta$ forward paths of the inner and outer control loops. The resulting simulation model structure is shown in Figure 10.

Since the OWA is usually skewed during the transition flight phase from hypersonic to supersonic speeds, the initial flight states are chosen as $H = 8$ km and $Ma = 0.8$, and the sample OWA is initially in the straight wing configuration. The wing skewing rate is chosen to be $\dot{\Lambda} = 3°/s$. According to the research results in reference [1], a skewing rate of $3°/s$ can satisfy the quasi-steady assumption. In this case, the closed-loop aircraft can be regarded as a linear time-invariant system, and the corresponding relationship between the stability margins in Equations (29) and (31) also applies accordingly.

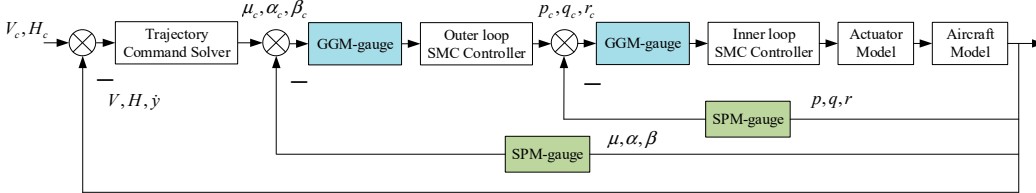

**Figure 10.** Structure of closed-loop aircraft simulation model containing both SPM-gauge and GGM-gauge.

When the oblique wing rotates from $0°$ to $30°$ at a rate of $3°/s$, the response curves of the system in Figure 10 under zero perturbation "$\varepsilon = 0, k = 1$", critical perturbation "$\varepsilon = 0.414, k = 3.162$", and beyond critical perturbation "$\varepsilon = 0.42, k = 3.2$" are plotted, as shown in Figure 11.

As shown in Figure 11, when no perturbation is imposed on the closed-loop aircraft, the flight altitude deviation is less than $7 \, m$, and the flight speed deviation is less than $1.6 \, m/s$, indicating that the designed reaching law index and error integral gain can ensure trajectory stability during the configuration transition process. When critical perturbation that can theoretically maintain system stability is imposed on this closed-loop aircraft, both the outer loop state variables $\mu, \alpha, \beta$ and the inner loop state variables $p, q, r$ generate a certain degree of oscillation based on zero-perturbation response, but this oscillation can eventually decay and disappear within a finite time. When perturbation that exceeds the critical stable value is imposed on this closed-loop aircraft, both the outer and inner loop state variables and the deflection angles of each control surface generate intense oscillations on the basis of the zero-perturbation response, and the motion of the closed-loop aircraft shows a divergent trend.

The above simulation results indicate that under the design values of $\omega_i = 0.24$ and $c_i = 1.47$, the actual SPM of the six control loops are all between $\varepsilon^* \in [0.414, 0.42]$, and the GGM $k_{\max}$ are all between $k_{\max} \in [3.162, 3.2]$, meeting the design requirements of $\varepsilon^* \geq 0.414$ in Equation (30) and $k_{\max} \geq 3.162$ in Equation (32).

Similarly, to verify that the GGM $k_{\min}$ of this closed-loop aircraft can meet the design requirements, the response curves of the system in Figure 10 under zero perturbation "$\varepsilon = 0, k = 1$", critical perturbation "$\varepsilon = 0.414, k = 0.263$", and beyond critical perturbation "$\varepsilon = 0.42, k = 0.25$" are plotted under the same wing skewing process, as shown in Figure 12.

Figure 12 shows that when critical perturbation is applied to the closed-loop aircraft, both the inner and outer loop state variables and the deflection angles of each control surface generate oscillations on the basis of the zero-perturbation response. However, this oscillation can decay and disappear within a finite time, and the response of the closed-loop aircraft still converges. When the singular perturbation increases from a critical value of 0.414 to 0.42, and the regularization perturbation decreases from a critical value of 0.263 to 0.25, due to the large deviation between the tracking error signal received by the sliding mode controller and its true value, precise control commands cannot be generated. Therefore, the above state variables exhibit increasingly high-frequency oscillations, and the response of closed-loop aircraft shows a divergent trend.

The above simulation results indicate that under the design values of $\omega_i = 0.24$ and $c_i = 1.47$, the actual SPM of the six control loops are all between $\varepsilon^* \in [0.414, 0.42]$, and the GGM $k_{\min}$ are all between $k_{\min} \in (0.25, 0.263]$, meeting the design requirements of $\varepsilon^* \geq 0.414$ in Equation (30) and $k_{\min} \leq 0.316$ in Equation (32).

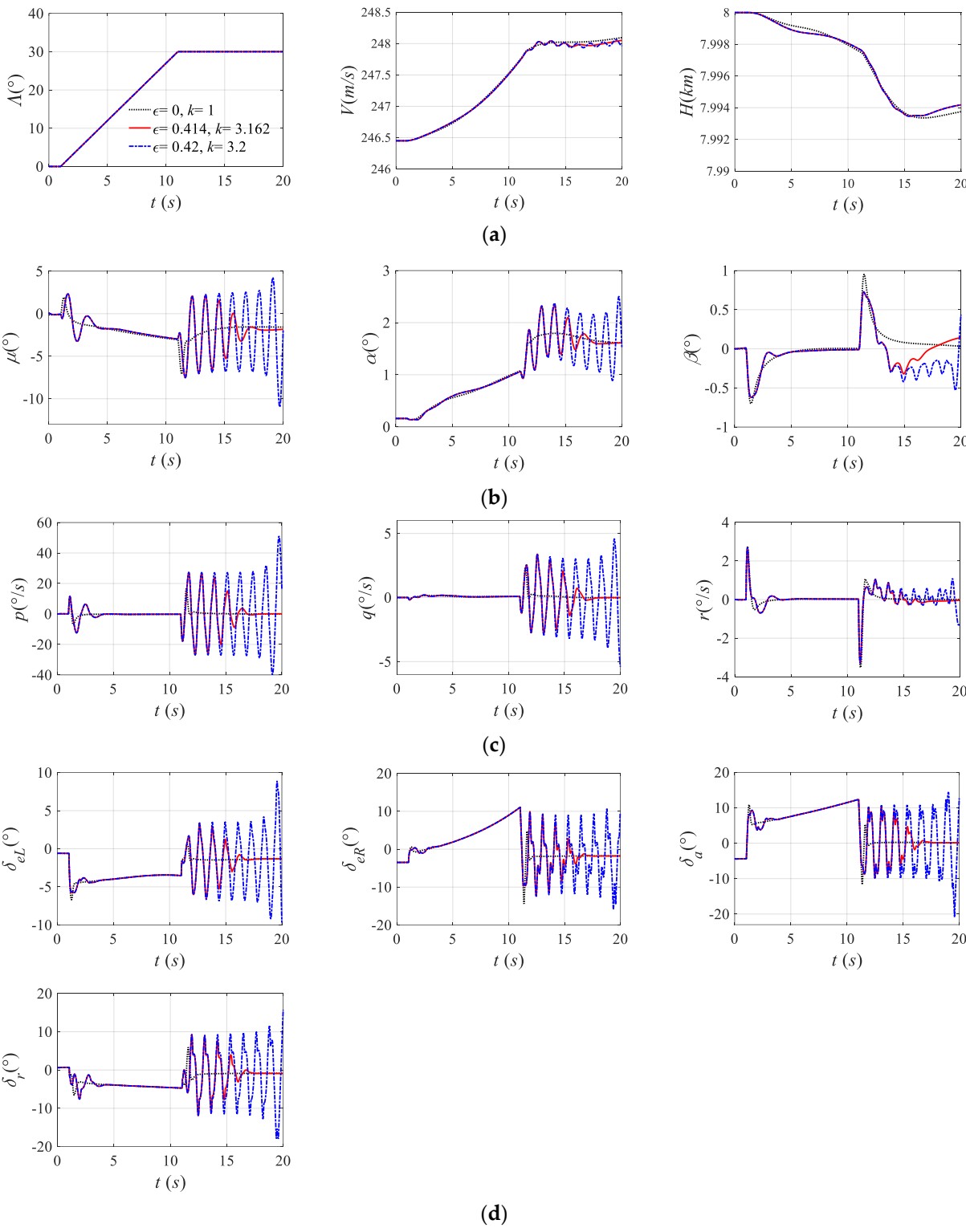

**Figure 11.** Verification curves of $\varepsilon^*$ and $k_{max}$ for closed-loop aircraft: (**a**) skew angle and flight states; (**b**) outer loop state variables; (**c**) inner loop state variables; (**d**) control surface deflection angles.

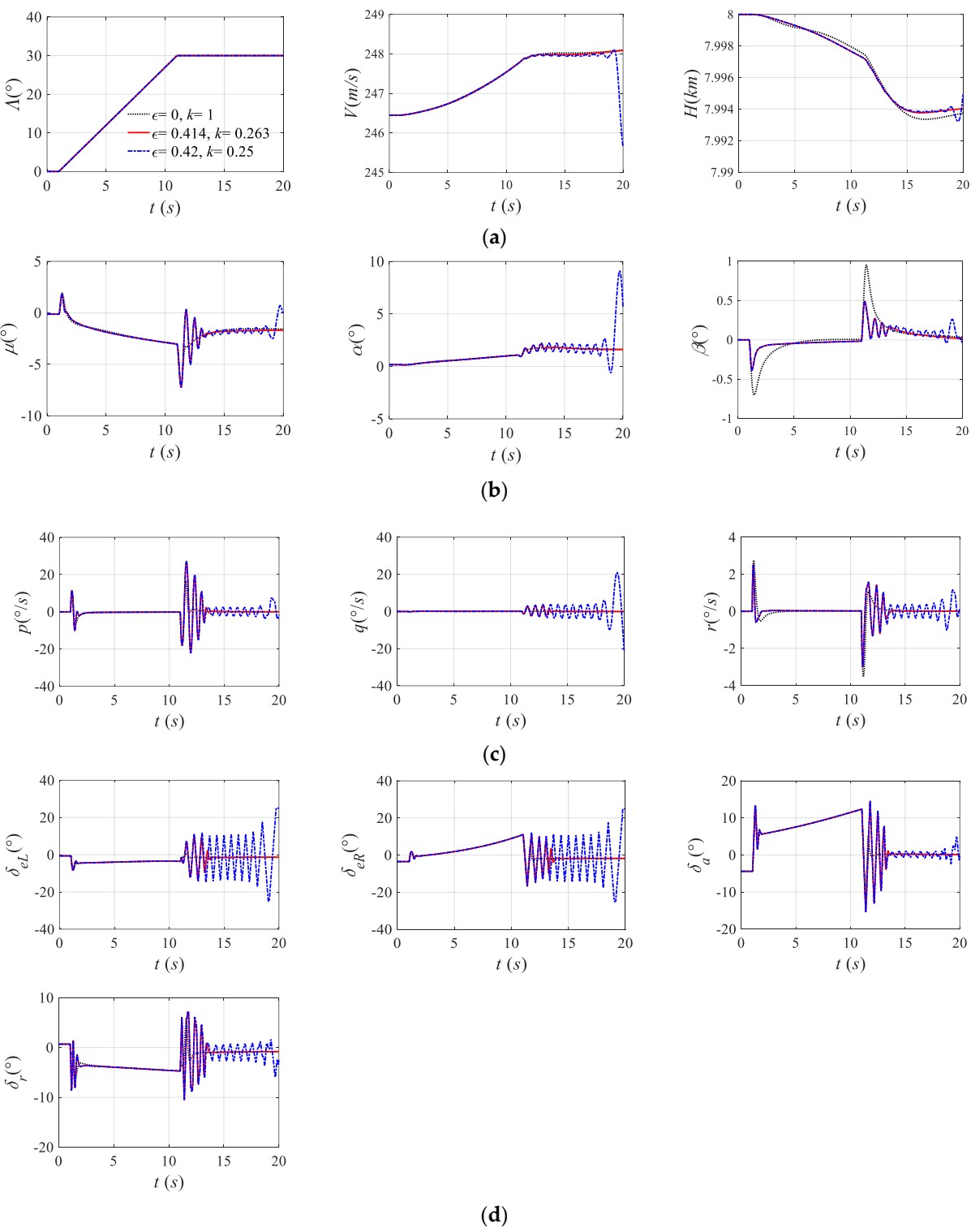

**Figure 12.** Verification curves of $\varepsilon^*$ and $k_{\min}$ for closed-loop aircraft: (**a**) skew angle and flight states; (**b**) outer loop state variables; (**c**) inner loop state variables; (**d**) control surface deflection angles.

## 7. Conclusions

The main work of this paper is to propose sliding mode flight control law design requirements for OWA based on perturbation theory. These design requirements represent the constraints on parameter values of the sliding mode control law, considering stability margin and command tracking speed, which can help designers select a combination of control law parameter values that can make the closed-loop aircraft have appropriate

stability margin, faster response speed, and faster error convergence speed from a variety of control law parameter values that only meet the accuracy of flight command tracking. The proposal of this design requirement can provide a theoretical basis for the values of control law parameters, greatly shortening the design cycle of the sliding mode controller.

(1) The composition and function of the sliding mode flight control structure of an OWA were introduced, and 12 design parameters of the sliding mode flight control law, including 6 reaching law indices and 6 error integral gains, were summarized. In the closed-loop aircraft model composed of an OWA and sliding mode controller, SPM-gauge and GGM-gauge were added, respectively, and the expressions for SPM and GGM were derived to establish the correspondence between the sliding mode control law parameters and the stability margin of this closed-loop aircraft model. The derivation results show that for a given SPM, the reaching law index and error integral gain form a closed curve in the first quadrant. For a given GGM, the reaching law index and error integral gain form three nonclosed curves with a monotonic decreasing trend in the first quadrant.

(2) Based on the quasi-steady assumption, the closed-loop aircraft model with a small wing skewing rate was regarded as a linear time-invariant system. The design requirements for the PM and GM of the linear time-invariant system were converted into limiting requirements for the SPM and GGM, respectively, and the value intervals of the sliding mode control law parameters were obtained. To ensure that the PM satisfies $|PM| \geq 45°$, the design requirement for the SPM should be $\varepsilon^* \geq 0.414$. To ensure that the GM satisfies $|GM| \geq 10$ dB, the design requirement for the GGM should be $k_{max} \geq 3.162$ and $0 < k_{min} \leq 0.316$. The value range of the sliding mode control law that meets the stability margin requirement is a two-dimensional closed region surrounded by curves $\varepsilon^* \geq 0.414$, $k_{max} \geq 3.162$, and $4n_1 - n_3 < 0$, where $4n_1 - n_3 < 0$ is a prerequisite for the existence of GGM.

(3) With the goal of reducing the sum of the reaching time and sliding time, the design values of the reaching law index and error integral gain were determined within the above value intervals. The design value of the reaching law index is recommended to be selected within the interval [0.24, 0.64], and the design value of the error integral gain is recommended to be selected within the interval [1.4, 2.1].

(4) The simulation results of the wing skewing process show that when the design values of the six reaching law indices are all selected as 0.24 and the design values of the six error integral gains are all selected as 1.47, the sample OWA can maintain trajectory stability during the wing skewing process. At this moment, the actual SPM of the six control loops are all between $\varepsilon^* \in [0.414, 0.42)$, and the GGM $k_{max}, k_{min}$ are all between $k_{max} \in [3.162, 3.2)$, $k_{min} \in (0.25, 0.263]$, meeting the design requirements of $\varepsilon^* \geq 0.414$, $k_{max} \geq 3.162$ and $0 < k_{min} \leq 0.316$.

**Author Contributions:** Conceptualization, L.W. and X.S.; methodology, L.W.; Software, X.S.; validation, L.W., X.S. and H.L.; formal analysis, S.T. and T.Y.; investigation, W.C. and S.T.; resources, J.M. and Y.Z.; data curation, J.M.; Writing—original draft, X.S. and H.L.; writing—review and editing, J.M. and W.C.; project administration, L.W. All authors have read and agreed to the published version of the manuscript.

**Funding:** This research received no external funding.

**Data Availability Statement:** Data are contained within the article.

**Acknowledgments:** The authors would like to deliver their sincere thanks to the editors and anonymous reviewers.

**Conflicts of Interest:** Authors Jingzhong Ma and Yun Zhu were employed by AVIC Jiangxi Hongdu Aviation Industry Group Company Ltd. The remaining authors declare that the research was conducted in the absence of any commercial or financial relationships that could be construed as a potential conflict of interest. The authors declare no conflicts of interest.

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
