# Peer review of "Sliding Mode Flight Control Law Design Requirements for Oblique Wing Aircraft Based on Perturbation Theory"

_aerospace, doi:10.3390/aerospace11050366_

Round 1

Reviewer 1 Report

Comments and Suggestions for Authors

As part of the peer review process for the manuscript titled "Sliding Mode Flight Control Law Design Requirements for Oblique Wing Aircraft Based on Perturbation Theory," I have carefully examined the content with an emphasis on its technical rigor, originality, and contribution to the field of aerospace engineering. The following comments are intended to guide the authors in refining their paper to meet the high standards of publication. Each suggestion aims to enhance the clarity, depth, and impact of the research presented.

This review addresses several areas in the manuscript that could benefit from additional attention:

1. Figure Consolidation: There appears to be considerable overlap between Figure 6 and Figure 8. Combining these into a single figure could improve the clarity and conciseness of the presentation, making it easier for readers to understand the visual data supporting your findings.

2. Inclusion of Mathematical Model: The manuscript does not currently present a clear mathematical model for the aircraft under study. Including such a model is crucial, especially given the technical nature of the topic and the need for thoroughness in demonstrating the application of perturbation theory in the design of the flight control laws.

3. Clarification of Novelty: The paper would benefit from a more explicit statement of its novel contributions, particularly in how it advances the application of sliding mode control to oblique wing aircraft. Highlighting these unique aspects would help delineate the paper’s significance within the broader field of aerospace engineering.

4. Enhancement of Discussion and Conclusion: Both the discussion and conclusion sections need strengthening to better encapsulate the research findings and their implications. It is important to more explicitly connect these sections to the study’s objectives, thereby solidifying the narrative from introduction to conclusion.

These recommendations aim to elevate the manuscript’s impact and its contribution to the field of flight control systems for oblique wing aircraft.

Reviewer 2 Report

Comments and Suggestions for Authors

1. The use of sliding mode controllers has several disadvantages for practical use, primarily including the appearance of chattering due to the influence of unmodeled dynamics, sampling of the control process, and emphasizing the influence of measurement noise. Therefore, their use is advisable in cases in which their usefulness is undoubted - especially due to significant uncertainty in the dynamics of the object and external disturbances. I believe that the article should very convincingly demonstrate the need and advantages of sliding modes for the control problem under consideration.

2. "during the wing skewing process" - please clarify, if this process is considered quasi-time-invariant (in comparison with transients of the state variables), or if it is essentially time-varying.

3. Eq. (1) needs to be clarified:  Eq. (1b) is the autonomous ODE, providing the function s(t) for a given s(0). At the same time, Eq. (1a) gives s(t) depending on e(t). Which one s(t) should be used?

4. The authors assume the possibility of directly measuring all state variables, which is usually impossible in practice.

5. It is unclear which aircraft dynamics model was used for calculations, which makes it impossible to reproduce the results.

6. Neither the theory nor the simulations take into account the dynamics of actuators and noise (errors) of measurements, which is important when using sliding modes.

Editing remarks.

1. "During oblique" - "During the oblique"

2. "specified sliding mode" - maybe: "specified sliding surface"?

3. "In recent years, Yang X et al." - please specify in [] the reference number

4. "Cheng, L et al. designed" -  the same, and throughout the paper.

5. Please rephrase "consists of two parts: a sliding surface" - "a sliding surface" is a surface, not a part of the control law

Comments on the Quality of English Language

"During oblique" - "During the oblique"

In general, English is satisfactory

Reviewer 3 Report

Comments and Suggestions for Authors

Authors analyzed how flight control law parameters influence closed-loop aircraft stability margins and proposed design requirements for sliding mode flight control laws. It introduced the composition and function of the sliding mode flight control structure, summarizing 12 design parameters. By adding SPM-gauge and GGM-gauge to the closed-loop aircraft model, expressions for Stability Prediction Margin (SPM) and Gain Margin (GGM) were derived. Under the quasi steady assumption, the closed-loop aircraft model was treated as a linear system, with design requirements for Phase Margin (PM) and Gain Margin (GM) translated into limits for SPM and GGM. Simulation results demonstrated that selected parameters maintained stability margins and trajectory stability during wing skewing.

The paper is well-suited to the journal's scope, and I found it interesting. Although the text is generally well-written, I believe that making some minor enhancements could further improve its fluency in reading and enhance the clarity of the message. There are a few issues for which I recommend a review:

1. The introduction clearly outlines the content of each section of the article. However, it lacks indication of the novel elements present in this paper. Please explicitly underline the main novelty of the paper in a single sentence at the end of the introduction.

2. Increasing the font size in some diagrams (e.g. Figures 6, 7, 8), would significantly improve their readability. The legend in these drawings is completely illegible

3. In some cases (Figures 12, 13) the font is so small that the charts become completely unreadable

4. To facilitate comprehension, it could be advantageous to organize all the quantities utilized in the equations into a separate paragraph. Doing so would not only improve the accessibility of the work but also aid readers in understanding the various components involved in the calculations.

At the moment, I've provided all the feedback I have. My intention is that these comments will offer valuable insights to the authors, facilitating enhancements to their paper.

Comments on the Quality of English Language

Although the text is generally well-written, I believe that making some minor enhancements could further improve its fluency in reading and enhance the clarity of the message. 

Reviewer 4 Report

Comments and Suggestions for Authors

This paper describes a sliding mode flight control structure and the design parameters of the control laws for oblique wing aircraft based on the singular perturbation margin and generalized gain margin. The parameters are applied to an example oblique wing aircraft. The article is well written and easy to follow. I would recommend publication after addressing a minor point. I would like more information about the aircraft that is described in the text, where it was referenced from, and what the geometry of the control surfaces are. I also think it would be useful for the authors to describe the implications of how the results can be applied to other oblique wing aircraft of different scales.  

In addition, I recommend making the following formatting changes in the text:

Line 127: is this a flight control structure for an example, general OWA? The language doesn't make it clear if this is for a specific type of OWA or a general case.

Figure 1, 9: please give a reference for this figure if not your own

Figure 6, 7, 8, 12, 13: Please increase the font size to a minimum of 8 pt

Round 2

Reviewer 1 Report

Comments and Suggestions for Authors

It can be accept in present form.

Reviewer 2 Report

Comments and Suggestions for Authors

The authors were attentive to all my comments and suggestions, significantly improving the manuscript. There are no more criticisms on my part.

Comments on the Quality of English Language

There are no more criticisms on my part.